# Development of VLA4 and CXCR4 Antagonists for the Mobilization of Hematopoietic Stem and Progenitor Cells

**DOI:** 10.3390/biom14081003

**Published:** 2024-08-14

**Authors:** Peter G. Ruminski, Michael P. Rettig, John F. DiPersio

**Affiliations:** Division of Oncology, Department of Medicine, Washington University School of Medicine, 660 S. Euclid Ave., St Louis, MO 63105, USA

**Keywords:** CXCR4, VLA-4, stem cell mobilization

## Abstract

The treatment of patients diagnosed with hematologic malignancies typically includes hematopoietic stem cell transplantation (HSCT) as part of a therapeutic standard of care. The primary graft source of hematopoietic stem and progenitor cells (HSPCs) for HSCT is mobilized from the bone marrow into the peripheral blood of allogeneic donors or patients. More recently, these mobilized HSPCs have also been the source for gene editing strategies to treat diseases such as sickle-cell anemia. For a HSCT to be successful, it requires the infusion of a sufficient number of HSPCs that are capable of adequate homing to the bone marrow niche and the subsequent regeneration of stable trilineage hematopoiesis in a timely manner. Granulocyte-colony-stimulating factor (G-CSF) is currently the most frequently used agent for HSPC mobilization. However, it requires five or more daily infusions to produce an adequate number of HSPCs and the use of G-CSF alone often results in suboptimal stem cell yields in a significant number of patients. Furthermore, there are several undesirable side effects associated with G-CSF, and it is contraindicated for use in sickle-cell anemia patients, where it has been linked to serious vaso-occlusive and thrombotic events. The chemokine receptor CXCR4 and the cell surface integrin α4β1 (very late antigen 4 (VLA4)) are both involved in the homing and retention of HSPCs within the bone marrow microenvironment. Preclinical and/or clinical studies have shown that targeted disruption of the interaction of the CXCR4 or VLA4 receptors with their endogenous ligands within the bone marrow niche results in the rapid and reversible mobilization of HSPCs into the peripheral circulation and is synergistic when combined with G-CSF. In this review, we discuss the roles CXCR4 and VLA4 play in bone marrow homing and retention and will summarize more recent development of small-molecule CXCR4 and VLA4 inhibitors that, when combined, can synergistically improve the magnitude, quality and convenience of HSPC mobilization for stem cell transplantation and ex vivo gene therapy after the administration of just a single dose. This optimized regimen has the potential to afford a superior alternative to G-CSF for HSPC mobilization.

## 1. Introduction

### 1.1. Indications for HSPC Mobilization

Hematopoietic stem and progenitor cells (HSPCs) are required for both autologous and allogeneic hematopoietic stem cell transplantation (HSCT) and, more recently, HSPC-based gene therapies being developed to treat genetic diseases such as sickle-cell anemia [1,2]. The phenomenon of HSPC mobilization, which was discovered quite by chance in patients recovering from chemotherapy almost 45 years ago [3], has transformed the clinical practice of HSPC transplantation [4]. HSCT and the more recent stem-cell–based gene therapies rely on the ability to harvest a sufficient number of CD34^+^ HSPCs from the patient (autologous) or from an HLA-matched or mismatched donor (allogeneic), typically via mobilization of these HSPCs from the bone marrow (BM) into the peripheral blood. Successful autologous and allogeneic HSCT, as well as gene therapy for blood borne diseases, requires the infusion of these mobilized HSPCs, collected by a process called apheresis, that are then capable of homing to the BM and regenerating durable trilineage hematopoiesis in a reasonable timeframe.

In addition to HSCT and gene therapy, HSPC mobilizing regimens have been tested for their capacity to enhance the efficacy of chemotherapy against both hematological and non-hematological malignancies. Like HSPCs, several malignant cells rely on CXCR4 and VLA4 for their homing and retention within the BM and/or tumor microenvironments [5,6,7]. These interactions between malignant cells and their microenvironments provide potent protection against both spontaneous apoptosis and chemotherapy [8,9,10]. Targeted disruption of the CXCR4-CXCL12 and VLA4-VCAM1 axes with CXCR4 or VLA4 antagonists has been explored to enhance the efficacy of chemotherapy against both hematological (acute myeloid leukemia, acute and chronic lymphoid leukemia, and multiple myeloma) and non-hematological malignancies. Since these chemosensitization studies have been extensively reviewed elsewhere [11,12,13,14,15,16], this review will focus on the roles of CXCR4 and VLA4 in HSPC mobilization for HSCT and gene therapy applications.

### 1.2. Biology and Spatial Properties of the HSPC Niche

The majority of adult HSPCs reside in unique, specialized microenvironments of the BM which consist of marrow stromal cells, osteoblasts, osteoclasts and other extracellular matrix proteins such as collagens, fibronectins, proteoglycans [17,18,19,20]. HSPCs express a number of cell-surface molecules—such as CD34, very late antigen 4 (VLA-4), CXCR4, CXCR2, CD44, CD62L, lymphocyte-function-associated antigen-1, CD117 (c-kit), and Robo4—that mediate their adherence in the BM microenvironment (Figure 1) [17,19,21,22].

These interactions are important for regulating HSPC trafficking, in addition to self-renewal, proliferation and differentiation processes [19,23]. Only a small fraction of adult HSPCs can normally be found in the peripheral circulation at any given time [24,25,26]. Although random leakiness of HSPCs could be a possible factor for the presence of a small number of them in the peripheral blood, the regularity of this physiological HSPC egress from the BM niche would imply that there is a biological role involved [27,28,29]. The number of HSPCs in the peripheral circulation at steady state can be substantially affected by a variety of endogenous and exogenous factors, such as growth factors [30,31,32,33,34,35,36,37], chemotherapy [3,38,39,40], chemokines [41,42,43,44], chemokine and integrin receptor agonists and antagonists [45,46,47,48], bioactive lipids [49,50], exercise [51,52], infection, and inflammation [53,54]. This enforced egress of HSPCs into peripheral blood is referred to as mobilization. While the function of baseline circulating HSPCs remains somewhat unclear, pharmacologically induced HSPC mobilization is the preferred method for generating grafts for HSPC transplantation. This has essentially replaced traditional bone marrow transplants as the only therapeutically effective option for many hematopoietic malignancies as well as for some non-malignant diseases. HSCT normally requires the intravenous infusion of a minimum of 2 × 10^6^ CD34^+^ stem cells/kg recipient body weight. However, a dose of 5 × 10^6^ CD34^+^ cells/kg is the preferable number for early, consistent, and long-term multilineage engraftment [55,56,57]. If there is a failure or a delay in collecting sufficient HSPCs to proceed to transplantation, it often extends the time of high-dose chemotherapy, which increases the risk of disease progression in cancer patients.

### 1.3. Clinical Application of HSPC Mobilization

Present clinical practice for HSPC mobilization is achieved via dosing with an extended course of G-CSF, typically 5 or more days of daily injections. G-CSF-based HSPC mobilization regimens are currently the preferred graft source for virtually all autologous and the majority of allogeneic HSCTs owing to their generally higher stem cell contents, reduced rates of graft failure, and better overall survival compared to BM transplantation [3,58,59]. In most healthy donors and in patients undergoing autologous stem cell mobilization, G-CSF induces an average 50–100-fold increase in circulating HSPCs after 5 daily subcutaneous (SC) injections of 10 ug/kg of G-CSF [60,61]. Mechanistically G-CSF induces egress of HSPCs from the BM into the peripheral blood by expanding the stem cell mass in the BM and by modulating two of the major pathways involved in stem cell BM retention: the chemokine receptor CXCR4-CXCL12 and integrin VLA4-VCAM1 axes [62,63,64]. Attenuation of these pathways via G-CSF is achieved through downregulation of CXCL12 (SDF-1) expression in BM stromal cells and osteoblasts and through the activation of proteolytic pathways resulting in cleavage of critical tethers that form the ligand-receptor pairs expressed in stem cells and BM niche constituents [62,63,64,65,66,67]. While the role of specific proteases involved in this proteolytic cleavage of these key tethers remains unresolved [68,69], the cell surface protease dipeptidyl peptidase 4 (DPP-4, CD26), which cleaves and inactivates the CXCR4 ligand CXCL12, has been shown to be an essential mechanism for G-CSF-induced mobilization [70,71].

### 1.4. Clinical Limitations of Current HSPC Mobilization Platforms

Current practices for harvesting HSPCs with G-CSF for HSCT are costly, involve a multi-day procedure with suboptimal stem cell yields in up to 30% of patients and are associated with some morbidity, including significant bone pain and on rare occasions splenic rupture, myocardial infarction, cerebral ischaemia and vaso-occlusive episodes (in sickle-cell patients) [72,73,74,75,76,77]. G-CSF is therefore often combined with the FDA approved small-molecule CXCR4 inhibitor plerixafor (AMD3100; Sanofi, Paris, France) to boost the number of HSPCs collected. Unfortunately, up to 24% of patients receiving plerixafor and G-CSF still fail to collect enough CD34^+^ HSPCs in 4 days of apheresis [78,79]. Likewise, plerixafor or (BL-8040; BioLineRx, Modi’in-Maccabim-Re’ut, Israel), an alternative high-affinity CXCR4 inhibitor, are too weak for efficient clinical mobilization when given as single agents to normal healthy donors (~33% mobilization failure rate) [80,81,82]. Furthermore, these regimens frequently yield suboptimal CD34^+^ HSPC numbers for HSPC-based gene-edited therapies, given the significantly higher HSPC number needed for successful gene editing and manufacturing [83,84,85]. 

The shortcomings and the inherent disadvantages of G-CSF, such as the slow mode of action [36,86], side effects, and associated contraindications [87,88], as well as significant heterogeneity in the mobilization response among donors [61], have driven efforts to identify alternative HSPC mobilization regimens. These new strategies would help address patients who fail initial mobilization, decrease the number of leukaphereses required to collect an adequate number of HSPCs, improve immune reconstitution, and presumably decrease total overall cost.

As indicated above, both the chemokine receptor CXCR4 and the cell-surface integrin receptor α4β1 (VLA4) are essential for the homing and retention of HSPCs within the bone marrow microenvironment. Preclinical and/or clinical studies have demonstrated that disrupting the interaction of CXCR4 or VLA-4 with their endogenous ligands can result in the rapid and reversible mobilization of HSPCs into the peripheral circulation within hours, as opposed to the multi-day process required with G-CSF [2]. Furthermore, the simultaneous inhibition of both VLA4 and CXCR4 results in an additive or even synergistic mobilization effect, rapidly mobilizing HSPCs within an hour in mice [2].

Consequently, it would seem advantageous to identify and optimize inhibitors of both CXCR4 and VLA4 to use in tandem as a rapid and more favorable HSPC mobilizing regimen as an alternative to G-CSF.

## 2. HSPC Mobilizing Agents That Target CXCR4

CXCR4 belongs to a large family of seven-transmembrane domain receptors coupled to heterotrimeric G proteins. It is expressed as different isoforms, via differential splicing, which affect the length of its N terminus [89]. The binding of CXCR4 to its endogenous ligand CXCL12 (SDF-1) results in the activation of multiple signal transduction pathways, ultimately triggering chemotaxis [89,90]. The CXCR4-CXCL12 signaling axis also plays a key role in maintaining HSPC self-renewal and quiescence [91,92,93,94]. In addition to CXCL12, both trefoil factor family 2 [95] and macrophage migration-inhibiting factor [96] were described as additional ligands capable of binding to CXCR4. CXCR4 expression on human CD34^+^ stem cells is dynamic, with the Flt3-ligand [97], SCF [98], IL-3 [99], IL-8 [100], hepatocyte growth factor [101], and G-CSF [62] all showing an ability to modulate CXCR4 expression and its pathways. 

A variety of different CXCR4 modulators, including small-molecule antagonists, peptide agonists and anti-CXCR4 antibodies have been reported previously, targeting a number of therapeutic indications [102,103,104,105,106]. This more recently includes the orally active small molecule mavorixafor (X4 Pharmaceuticals, Boston, MA, USA), which was FDA approved as a treatment for the rare disease WHIM (warts, hypogammaglobulinemia, infections, and myelokathexis) syndrome [107].

Several preclinical and clinical studies have previously demonstrated that pharmacological disruption of the CXCL12/CXCR4 axis using CXCR4 inhibitors can stimulate HSPC mobilization in a rapid and target-dependent manner [36,108]. 

We previously described in detail the roles that CXCR4 and VLA4 play in HSPC retention in the bone marrow and how inhibition of these receptors can lead to rapid mobilization of HSPCs into the peripheral blood [2]. At that time, plerixafor, a small-molecule bicyclam CXCR4 antagonist, was the only CXCR4 inhibitor approved by the FDA in association with G-CSF for autologous stem cell mobilization in patients with non-Hodgkin’s lymphoma and multiple myeloma (MM) [78,79]. When used alone, plerixafor is a rapid, but relatively weak mobilizer of HSPCs with a somewhat short duration of action [82]. When combined with G-CSF, however, plerixafor increases peripheral CD34^+^ concentration 2–3-fold compared to G-CSF alone [78,79]. Despite this, up to 24% of patients undergoing autologous stem cell transplantation for the treatment of lymphoma who receive plerixafor and G-CSF still fail to collect ≥2 × 10^6^ CD34^+^ cells/kg in over 4 days of apheresis [78,79].

Since that prior review article [2], motixafortide—a novel, small-molecule, cyclic-peptide-based CXCR4 inhibitor—was further developed and is reported to have extended in vivo activity (>48 h) in humans. Motixafortide was recently approved by the FDA in September 2023 [109] based on evidence from the GENESIS study, a double-blind, placebo-controlled prospective randomized study in which 122 participants with multiple myeloma undergoing autologous transplantation were randomized 2:1 to receive motixafortide 1.25 mg/kg with G-CSF or a placebo with G-CSF for mobilization of HSPCs for collection and apheresis [110]. Motixafortide plus G-CSF enabled 92.5% of patients to collect ≥6 × 10^6^ CD34^+^ HSPCs/kg recipient body weight within two apheresis procedures (primary endpoint) whereas only 26.2% of participants treated with placebo plus G-CSF achieved this mobilization yield [110]. Furthermore, the data suggest that motixafortide preferentially mobilizes increased numbers of more primitive HSPCs, as shown by immunophenotyping and single-cell RNA expression profiling [110].

## 3. HSPC Mobilizing Agents That Target VLA-4

### 3.1. The Alpha 4 Integrin Family

There are two α4 integrins, α4β1 (VLA4) [111] and α4β7 [112]. The α4β1 integrin normally exists in the inactive (low-affinity) state. Although the precise mechanism of activation in vivo is not clear, VLA4 can be activated in vitro via interaction with a ligand, divalent cations [113,114,115], monoclonal antibodies [113,114,115], CXCL12 [116,117,118,119], IL-3 [120], c-kit ligand [120,121], GM-CSF [120], stem cell factor [122], and phorbol esters. Once activated, the main endogenous ligands for VLA4 are vascular cell adhesion molecule-1 (VCAM-1; CD106) [123,124] and the alternatively spliced connecting segment 1 domain of fibronectin [125,126] which is found in the extracellular matrix [127]. VLA-4 can also bind to mucosal addressin cell adhesion molecule 1 (MADCAM1), osteopontin, intercellular adhesion molecule-4 (ICAM4), thrombospondin, as well as to some metalloprotease family members, but it is not clear what the biological significance of these additional interactions might be [128]. MADCAM1 is the primary ligand for the α4β7 integrin [129,130]. The extracellular ligands VCAM-1 and MADCAM1 are both normally expressed in the gastrointestinal tract. However, VCAM-1 is also expressed within peripheral organs [131,132], whereas MADCAM1 expression is normally restricted to the gut [131,133,134]. VLA4 is constitutively expressed on most leukocytes, including monocytes, lymphocytes, eosinophils, basophils, and CD34^+^ HSPCs, including early erythroid progenitors. IL-3 and SCF have been shown to upregulate the expression of VLA4 on the surface of CD34^+^ HSPCs. Alternatively, G-CSF downregulates VLA4 expression and is therefore associated with mobilization of CD34^+^ HSPCs [135,136,137,138,139,140]. 

In vivo administration of anti-α4 integrin antibodies has been shown to increases the number of peripheral circulating HSPCs in mice, primates, and humans compared to untreated controls. Such inhibition in mice [45,46,141,142,143,144] and nonhuman primates [45,46,47] leads to a rapid and prolonged mobilization of HSPCs. Colony-forming unit cell (CFU-C) assays and competitive transplant studies in mice have shown that the anti-α4 integrin antibody can mobilize committed progenitors and long-term repopulating cells, provided the HSPCs have a functional c-kit receptor [142]. When the α4 antibody was combined with G-CSF [46,47], plerixafor [45], c-kit ligand [46], and/or Flt3-ligand [34], an additive or even synergistic HSPC mobilization effect was shown to occur. This enhanced mobilization of HSPCs is likely mediated through the disruption of the CXCR4/CXCL12 axis signaling in the bone marrow in addition to inhibition of VLA4 [145]. 

### 3.2. Small-Molecule VLA4 Inhibitors

There have been previous efforts to develop small-molecule inhibitors of the VLA-4 integrin receptor as an alternative to the FDA approved alpha-4 antibody Natalizumab (Biogen, Cambridge, MA, USA), which is used clinically for the treatment of multiple sclerosis (MS) and in Crohn’s disease [146]. For these indications, the VLA4 inhibition mechanism attenuates inflammatory lymphocyte trafficking into the CNS and GI tract, respectively.

We and others previously reported that BIO5192, a very potent and selective small-molecule inhibitor of VLA-4, with an affinity 250- to 1000-fold higher than for the related α4β7 integrin [48,147], or BOP, a dual α4β1/α9β1 inhibitor, can induce rapid and reversible mobilization of murine HSPCs into the peripheral blood when compared to vehicle control and exhibit a synergistic effect when combined with G-CSF and/or plerixafor [48,148]. The functional characteristics of these HSPCs mobilized into peripheral blood with subcutaneous administration of BIO5192 was also assessed in competitive long-term repopulating studies. Mice that received transplants of HSPCs mobilized after a single dose of BIO5192 plus plerixafor had a durable engraftment outcome that approached transplants of HSPCs mobilized via a 4-day course of G-CSF. Additionally, the combination of BIO5192 and plerixafor compared to BIO5192 or plerixafor alone led to higher levels of donor chimerism. Furthermore, secondary transplantation in mice demonstrated stable engraftment of BIO5192-mobilized cells, suggesting that BIO5192, like plerixafor and G-CSF, can mobilize primitive HSPCs capable of providing long-term multilineage engraftment [48]. 

Unfortunately, BIO5192 exhibits poor pharmacokinetic properties (very short plasma half-life). This is partially compensated for by its strong binding affinity to its VLA4 target (slow off rate), leading to an extended pharmacodynamic (PD) effect, but BIO5192 also has significant solubility issues that cause difficulty in formulation for subcutaneous dosing and thus was not advanced as a clinical candidate.

A number of other small-molecule VLA4 inhibitors have been previously reported [149,150], and several have advanced through various stages of human clinical trials, including firategrast, TR-14035, carotegrast methyl (AJM300; EA Pharma, Tokyo, Japan), and valategrast, which were being developed for multiple sclerosis, inflammatory GI indications, and/or asthma [149,150]. Only one, carotegrast methyl (an oral α4 integrin antagonist ester pro-drug), at very high doses (960 mg three times daily), has recently been approved in Japan for ulcerative colitis after completion of Phase III trials [151] (see Table 1). None of these were considered for HSPC mobilization, however.

Our lab’s evaluation of these previously reported small-molecule clinical VLA4 inhibitors has shown either poor pharmacokinetic (PK) or metabolic properties, insufficient VLA4 inhibition potency (thus requiring high doses to achieve in vivo efficacy) and/or very poor solubility, making them inadequate candidates to use for extended HSPC mobilization after a single subcutaneous or oral dose. Firategrast, for example, required oral dosing of 1200 mg (in men) and 900 mg (in women) twice daily in order to observe a positive biomarker readout in a Phase II multiple sclerosis trial [152], and further advancement of this drug was discontinued. As noted above, carotegrast methyl required oral dosing of 960 mg three times daily in order to see a positive effect in ulcerative colitis.

Some of these previously reported inhibitors, and some of our own lab’s early leads, can rapidly mobilize HSPCs into the peripheral blood of mice after a single subcutaneous injection (often at very high doses), but the effect is short-lived and provides an inadequate number of HSPCs necessary for stem cell transplantation or for gene therapy [153,154]. Therefore, we sought to develop potent VLA4 inhibitors with improved pharmacokinetic, pharmacodynamic, and physiochemical properties, imparting significant and extended HSPC mobilization (of at least 6 h) after a single dose. Once obtained, we envisioned that an optimized VLA4 inhibitor could then be administered as a single dose in combination with a CXCR4 antagonist, such as plerixafor or motixafortide, to provide an additive or synergistic HSPC mobilization effect lasting six hours or more that could be used clinically as an alternative to the multi-day dosing regimen now required of G-CSF to achieve the same level of HSPC mobilization.

We recently synthesized multiple iterations of VLA4 inhibitor molecules based on prior reported structure-activity relationship (SAR) studies and tested their VLA4 inhibition potency using soluble VCAM-1 binding assays [154]. The most potent of these new Inhibitors, and those with improved aqueous solubility, were then tested in mice for extended HSPC mobilization beyond 4 h after a single injection, both alone as well as in combination with the CXCR4 inhibitor plerixafor or motixafortide. HSPC mobilization was measured in wild-type and splenectomized mice via flow cytometry to quantify the proportion of LSK (Lineage- Sca+ cKit+) cells as well as via colony-forming unit (CFU) assays [154]. For competitive transplantation studies [154], mobilized CD45.1^+^ BALB/c mouse blood was injected into lethally irradiated CD45.2^+^ BALB/c recipients alongside CD45.2^+^ BALB/c bone marrow cells. Mobilized HSPC engraftment was monitored monthly via flow cytometry for CD45.1^+^ cells in peripheral blood.

Early leads from our lab’s initial efforts resulted in extremely potent (sub-nanomolar) inhibitors of VLA4 that facilitated rapid HSPC mobilization in mice after a single subcutaneous dose. However, as with other previous inhibitors, the mobilization effect was only sustained for ~2 h after dosing (Figure 2)—not long enough to collect an acceptable number of HSPCs to proceed to transplant or for gene therapy [153,154,155,156]. Most compounds also had poor solubility and very short plasma half-lives after subcutaneous injection in mice. Fortunately, we were then able to identify a targeted spot on our optimized scaffolds for covalent attachment that would tolerate polyethylene glycol (PEG) functional groups of increasing PEG chain lengths that still maintained sub-nanomolar inhibition in the soluble VCAM-1 binding assays, comparable to their non-pegylated parent analogues. These efforts ultimately led to the discovery of lead molecules, which provide optimal and extended mobilization in mice out to 6 h or greater (Figure 2) [155,156]. This coincided with improved PK properties (a more extended drug plasma concentration) and also significantly improved aqueous solubility, allowing for dissolution in plain saline.

These new lead inhibitors mobilize two-fold more murine HSPCs and for a longer period of time than previously published best-in-class VLA4 inhibitors or our own early VLA4 inhibitor leads [155]. Furthermore, our novel, long-acting PEG analogues demonstrate superior HSPC mobilization when compared to an alpha-4-targeted antibody (anti-CD49d) in mice [156] and, as observed previously, synergistically mobilize murine HSPCs upon co-administration with a CXCR4 inhibitor (plerixafor or motixafortide) [155,156]. A similar synergistic effect was also achieved in non-human primates when one of our leads was tested in combination with plerixafor [155]. A competitive repopulation study in mice demonstrated superior engraftment durability from HSPCs mobilized with our lead VLA4 inhibitors in combination with a CXCR4 inhibitor, especially motixafortide, when compared to HSPCs mobilized with G-CSF. This drug combination mobilized long-term repopulating cells that successfully engraft and expand in a multilineage fashion in secondary transplant experiments [133]. These newly optimized VLA4 inhibitors combined with a CXCR4 inhibitor, such as the recently approved motixafortide, provide encouragement that a safe and quicker-acting HSPC mobilization regimen as an alternative to G-CSF may be within reach.

## 4. Current and Future Trends of Stem Cell Mobilization

Modern stem cell mobilization for autologous stem cell transplantation for multiple myeloma has evolved over time. The use of chemotherapy plus G-CSF for mobilization was initially utilized to optimize stem cell yields, but G-CSF plus either plerixafor or motixafortide can mobilize sufficient HSPCs (>6 × 10^6^ CD34^+^/kg recipient body weight) for two transplants after 2–4 apheresis collections [79,110]. Additionally, the use of tandem autologous transplantation has largely disappeared from the normal clinical practice for myeloma since multiple other more effective and less toxic therapies are now available, including small molecules, bispecifics and chimeric antigen receptor (CAR)-T cells. In our center, only 2% of the patients with myeloma that underwent a single autotransplant were treated with a second autotransplant in the last five years. For potential cost-saving, many centers utilize prediction algorithms for timing of initiation of stem cell collection (usually >10 CD34^+^/uL blood) [157] and for determining if one or more doses of a CXCR4 inhibitor are necessary to achieve optimal stem cell mobilization.

The application of predictive algorithms provides a valuable tool to aid decision-making in various aspects of stem cell collection, including the timing of collection, procedure scheduling, processing requirements, the number of required procedures, procedure duration, and minimizing the storage of surplus product. These principles hold true for autologous stem cell mobilization and transplantation of patients with non-Hodgkin’s lymphoma, which is also less frequently offered as a therapeutic option again due to more effective and potentially less toxic immunotherapies. For patients undergoing allogeneic stem cell transplantation, G-CSF is the only approved treatment for mobilization and results in successful mobilization of >5 × 10^6^ CD34^+^/kg in a single apheresis in >70% of normal donors. In contrast, only 6–16% of normal donors receiving a single dose of plerixafor or motixafortide provide enough HSPCs in a single apheresis session to allow allogeneic recipients to proceed to transplant [158]. Some clinical trials are utilizing a G-CSF-plus-plerixafor regimen to maximize CD34^+^ HSPC collection for gene editing, base editing, and lentiviral transduction. However, since G-CSF is unsafe and contraindicated in patients with sickle-cell disease, robust and rapid stem cell mobilization regimens lacking G-CSF are needed for autologous gene therapy for patients with sickle cell anemia, which may require >20 × 10^6^ CD34^+^/kg. The use of novel approaches, such as motixafortide or motixafortide plus small-molecule VLA-4 inhibitors, may be the future of mobilization for this population of sickle cell anemia patients undergoing curative gene therapy and autologous stem cell transplantation.

## 5. Summary

Both the chemokine receptor CXCR4 and the integrin receptor VLA4 have important roles in the homing and retention of HSPCs within the bone marrow microenvironment. Disrupting the interaction of these receptors on the surface of the stem cells with their endogenous ligands within the bone marrow niche leads to their rapid egress into the peripheral blood. Targeting both receptors with pharmacological inhibitors has the potential to quickly drive a significant number of HSPCs into the peripheral blood, where a donor could undergo apheresis over several hours and potentially collect a sufficient number of CD34^+^ HSPCs to be used for transplant or gene therapy. Potent inhibitors of CXCR4 and VLA4, when given in combination, may provide a viable—and potentially safer, faster, and more cost-effective—alternative to G-CSF as a stem cell mobilization therapeutic regimen. With the potent, long-acting, and recently approved CXCR4 inhibitor motixafortide and the promising optimized small-molecule VLA4 inhibitors we are currently developing, such a combination HSPC mobilization therapeutic regimen may provide such an option in the near future.

## Figures and Tables

**Figure 1 biomolecules-14-01003-f001:**
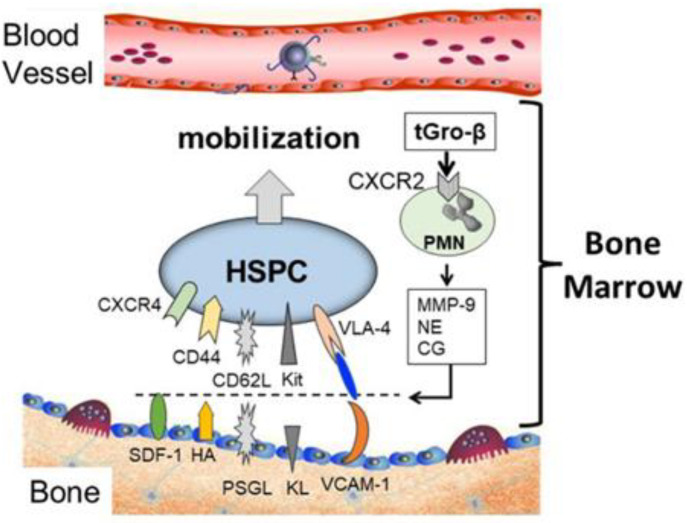
Representation of hematopoietic stem and progenitor cells (HSPCs) and the key receptors and ligands used for binding and retention in the bone marrow niche. Disruption of these binding interactions can lead to the HSPC mobilization into the peripheral blood.

**Figure 2 biomolecules-14-01003-f002:**
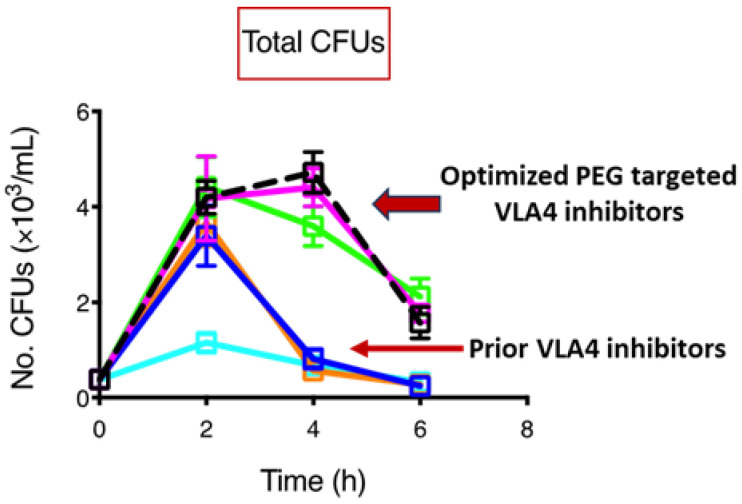
Graph showing the number of colony-forming units (CFUs) collected and measured (as per reference [138]) from the peripheral blood of mice out to 6 h. after a 3 mg/kg SC injection of representative examples of small-molecule VLA4 inhibitors. Prior VLA4 inhibitors represent early leads from our lab (depicted in the graph in blue, light blue and orange colors), which are potent inhibitors but have poor pharmacokinetic properties, leading to a rapid mobilization effect peaking at 2 h. However, values returned to near baseline by 4 h, which is not enough time to collect an adequate number of HSPCs for HSCT. The optimized VLA4 inhibitors with targeted PEG groups covalently attached to our hybrid core molecule (depicted in the graph in light green, magenta and dotted black colors) exhibit improved pharmacokinetics and solubility and provide for rapid mobilization after a single SC injection that maintains a peak out past 4 h and continue to maintain mobilized HSPCs at 6 h.

**Table 1 biomolecules-14-01003-t001:** Inhibitors of VLA4 that have been in human clinical trials.

Drug Name	Class	Target Indication	Furthest Phase	Status
Firategrast	Small Molecule	Multiple Sclerosis	Phase II	Discontinued
Valategrast	Small Molecule	Multiple Sclerosis/ Asthma	Phase II (asthma); Phase I (MS)	Discontinued
TR-14035	Small Molecule	Asthma/RA/ Multiple Sclerosis	Phase II (asthma, RA); Phase I (MS)	Discontinued
Carotegrast methyl (AJM300)	Small Molecule	Ulcerative Colitis (UC)	Phase III	Approved in Japan for UC
Natalizumab	Antibody (anti-α4)	Multiple Sclerosis/ Crohn’s Disease	Phase III	FDA approved MS/Crohn’s

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
