# Peer review of "Development of VLA4 and CXCR4 Antagonists for the Mobilization of Hematopoietic Stem and Progenitor Cells"

_biomolecules, 2024, doi:10.3390/biom14081003_

Round 1
Reviewer 1 Report
Comments and Suggestions for Authors
Very nice review on subject of VLA4 and CXCR4 antagonists for the mobilization of HPC/HSC.
One minor suggestion to improve content would be addition of a table listing at least some of the various CXCR4 and VLA4 antagonists they and others have developed and evaluated for mobilization that includes molecular class, pharmacological properties, and pharmacodynamic properties, including advantages/disadvantages and potential for clinical development.
Author Response
Reviewer 1.
Comments:
Very nice review on subject of VLA4 and CXCR4 antagonists for the mobilization of HPC/HSC.
Response: We thank the reviewer for his/her kind comments
One minor suggestion to improve content would be addition of a table listing at least some of the various CXCR4 and VLA4 antagonists they and others have developed and evaluated for mobilization that includes molecular class, pharmacological properties, and pharmacodynamic properties, including advantages/disadvantages and potential for clinical development.
Response: As recommended by Reviewer #3 we have generated a Table (Table 1) which reviews all known VLA-4 inhibitors in human clinical trials. Of note, none of the previously described VLA-4 inhibitors have been tested in clinical trials for HSPC mobilization or are being developed for stem cell mobilization. We did not include detailed information about our efforts and lead VLA-4 inhibitor compounds since all of this is awaiting publication. We appreciate the interest of Reviewer #3 in CXCR4i development. We feel that this represents a major alternative focus and outside the scope of this review. The development of oral and IV/SC CXCR4i may deserve its own review, of which many exist. There are only two approved CXCR4 inhibitors for HSPC mobilization (Plerixafor and Motixafortide) which we have discussed at length in the text and thus we did not feel that a separate table was indicated.
Reviewer 2 Report
Comments and Suggestions for Authors
1. At the beginning, I suggest to add a brief summary of indications, clinical efficiency, and the problems of HSCT.
2. Organize the introduction and set up subheadings.
3. Does CXCR4 blockade influence self-renew and proliferation of HSPC?
4. I suggest to remove 3.1, since it is irrelevant to the main point of paper.
5. In 3.3, as a review, it is better to profile all current VLA-4 inhibitors, rather than all in one sentence and only presenting the results from your own lab.
Reviewer 3 Report
Comments and Suggestions for Authors
Comments:
This is a very expansive review of the pathways of stem cell mobilization as currently practiced for clinical purposes.It draws from the authors deep experience not only in the use of previously described materials covering the two independent pathways of mobilization but additional contributions using new agents with higher potency and important clinical utility.
Οnly two minor comments:
a}where is stated that a4 integrin is present in most leucocytes should be also added that is present in erythroid cells,
b}one of the issues that may need further amplification is the following:the impression is given that what mechanistically involved in mobilization of many other unrelated situations i.e following treatment with expansive cytokines,,marrow regeneration post transplantation or post different stress pathways [mobilization with these were described well before the other two pathways where known],is some down regulation of the two known pathways.What these diverse conditions have in common is the disruption of BM eco-system and the relationships of stem cells with other cells and micro environmental cells with an effect beyond the described influence on CXCR4 and a4.
Round 2
Reviewer 2 Report
Comments and Suggestions for Authors
All my concerns have been well addressed. There's no more question. I suggest to publish this paper.